# Matrix-Bound Zolzoledronate Enhances the Biofilm Colonization of Hydroxyapatite: Effects on Osteonecrosis

**DOI:** 10.3390/antibiotics10111380

**Published:** 2021-11-11

**Authors:** Ranya Elsayed, Ahmed El-Awady, Christopher Cutler, Zoya Kurago, Mahmoud Elashiry, Christina Sun, Ryan Bloomquist, Mohamed M. Meghil, Mohammed E. Elsalanty

**Affiliations:** 1Dental College of Georgia, Augusta University, Augusta, GA 30912, USA; relsayed@augusta.edu (R.E.); aelawady@augusta.edu (A.E.-A.); ccutler@augusta.edu (C.C.); zkurago@augusta.edu (Z.K.); melashiry@augusta.edu (M.E.); csun@augusta.edu (C.S.); rbloomquist@augusta.edu (R.B.); mmeghil@augusta.edu (M.M.M.); 2College of Osteopathic Medicine of the Pacific, Western University of Health Sciences, Pomona, CA 91766, USA

**Keywords:** osteonecrosis, oral bacteria, biofilm, MRONJ, osteoporosis, bisphosphonates, anti-resorptive, zoledronates

## Abstract

(1) Background: The aim of this study was to test whether matrix-bound zoledronate (zol) molecules enhanced the oral biofilm colonization of a mineralized matrix, rendering the alveolar bone more susceptible to medication-related osteonecrosis of the jaw (MRONJ) following invasive dental procedures. (2) Methods: We tested the effect of matrix-bound zol on the growth and attachment of *Porphyromonas gingivalis* (Pg), *Fusobacterium nucleatum* (Fn) and *Actinomyces israelii* (Ai), and whether the nitrogen-containing component of zol contributed to such effect. The role of oral bacteria in the induction of osteonecrosis was then tested using an extra-oral bone defect model. (3) Results: The attachment of biofilm to hydroxyapatite discs increased when the discs were pre-treated with zol. Bacterial proliferation was not affected. Matrix-bound zol was more potent than non-nitrogen-containing etidronate in enhancing the colonization. Stimulation was dampened by pre-treating the bacteria with histidine. The delivery of oral biofilm to a tibial defect caused osteonecrosis in zol-treated rats. (4) Conclusions: We conclude that matrix-bound zol enhances the oral biofilm colonization of hydroxyapatite. This enhancement depended on the presence of the nitrogen-containing group. The oral biofilm rendered the extra-oral bone susceptible to medication-related osteonecrosis, suggesting that it has an important role in the induction of MRONJ.

## 1. Introduction

Osteonecrosis related to anti-resorptive medications is a phenomenon exclusive to the jawbone. However, the role of local factors, such as the oral microbiome, in rendering the alveolar bone uniquely susceptible is not well understood. The presence of biofilm was confirmed in bone specimens from patients with established medication-related osteonecrosis of the jaw (MRONJ) [1]. The bacterial species included a variety of Gram-positive and Gram-negative species, with the majority being anaerobes or facultative anaerobes. The species *Fusobacterium*, *Actinomyces*, *Bacillus*, *Staphylococcus*, *Streptococcus*, *Selenomonas* and *spirochetes* were prevalent. Similar results were confirmed in 55 clinical studies, involving 814 patients over a ten-year period from 2003 until 2013 whereby Actinomyces was most prevalent in 68.8% of cases. Other species included *Streptococcus*, *Staphylococcus*, *Klebsiella*, *Eikenella*, *Haemophilus*, *Fusobacterium*, and *Escherichia* [2].

Some reports suggested that MRONJ could result from bacterial-induced bone resorption [1,3,4]. Periodontitis has been identified as an important risk factor for the development of MRONJ [5,6], suggesting that periodontitis-induced inflammation and associated bacteria could contribute to MRONJ pathogenesis [5,7,8]. A cross-sectional study of patients on long-term bisphosphonate (BP) treatment with or without a history of MRONJ reported that the IgG titer against *P.gingivalis* and gingival crevicular fluid (GCF) level of IL-1b were significantly associated with osteonecrosis [9]. Data from animal models of periodontitis supported this finding as well [10], some showing the involvement of additional strains such as *Aggregatibacter actinomycetemcomitans* [11].

It has also been suggested that the presence of BP in the bone matrix could favor bacterial attachment and/or growth, contributing to the pathogenesis of MRONJ [3,12,13,14]. This would promote the adhesion of gram-positive strains, such as *actinomyces*, facilitating the adherence of other microbial species, forming a resistant biofilm and dysbiosis [13,14]. Invasive dental trauma may be a triggering factor, whereby exposed bone provides a medium for bacterial colonization [14,15]. Direct electrostatic interaction with the amino-cationic group of nitrogen-containing bisphosphonates was suggested as a mechanism for this enhanced attachment [12,13,14]. Another suggested mechanism was the interaction of the BP nitrogen domain with the microbial surface components recognizing adhesive matrix molecules (MSCRAMM) of gram-positive bacteria [12,13]. Finally, since the incidence of ONJ is highest in cancer patients receiving high doses of nitrogen-containing BP [16], it is plausible that immune disturbance in those patients could modulate the oral microbiome in a direction that increases susceptibility to osteonecrosis [17].

In this study, the hypothesis that matrix-bound zoledronate (zol) enhanced the colonization of oral bacteria, represented by three important strains: *Porphyromonas gingivalis* (Pg), *Fusobacterium nucleatum* (Fn) and *Actinomyces israelii* (Ai). We then tested the effect of the nitrogen-containing group of the bisphosphonate molecule on bacterial attachment. Finally, we investigated the effect of oral biofilm on the induction of osteonecrosis in an extra-oral bone defect in the tibia in an established MRONJ rat model [18].

## 2. Results

### 2.1. Zol Increased Attachment of Oral Bacterial Species to Mineralized Matrix

We tested the effect of zol on the attachment of *Fusobacterium nucleatum* (*F. nucleatum*) and *Actinomyces israelii* (*A. isrealii*) to dentin discs pre-treated with or without zol. These strains are the most predominant bacterial species associated with MRONJ lesions [1,19]. An increase in this attachment to zol-modified dentin discs was observed with *F. nucleatum* with zol concentrations 1 µM, 10 µM and 100 µM (*p* = 0.004, *p* < 0.0001, *p* < 0.0001, respectively) (Figure 1A,B). Similar results were observed with *A. israelii* (*p* = 0.0152; Figure 1C).

### 2.2. Increased Attachment of WT vs. MFB FimA-/Mfa1-P. gingivalis to Dentin Discs Pre-Treated with Zol

Periodontitis is a known risk factor for MRONJ. We tested whether matrix-bound zol enhances the attachment of an important periodontal pathogen, *P. gingivalis*, to dentin. Results showed a significant increase in the attachment of WT *P. gingivalis* to the zol-treated dentin discs compared to the control discs and the fimbria-mutant *P. gingivalis* (*p*-value < 0.001; Figure 1D). *P. gingivalis’* fimbriae have been shown to play a crucial role in its attachment and virulence [20], which was further confirmed by our results showing lack of attachment of the MFB double fimbriae mutant. On the other hand, the growth curve showed no significant effect of zol on the bacterial growth or proliferation, as shown by its OD660 nm readings (Figure 1E). We further tested whether zol enhanced the major and minor fimbriae gene expression in *P. gingivalis*. Results showed a relative increase in both major and minor fimbriae—*fimA* and *mfa1*, respectively—using gene expression analysis (Figure 1F). Taken together, our results suggest that *P. gingivalis*’ preferential attachment to the zol-treated mineralized matrix may have involved a zol-induced increase in fimbrial expression.

### 2.3. Role of Zol Nitrogen Group in the Enhanced Bacterial Attachment to Zol-Modified Mineralized Matrix

We then compared the attachment of *P.gingivalis* to a dentin matrix treated with nitrogen and non-nitrogen-containing bisphosphonates (zol and etidronate, respectively). While the bacterial attachment was increased with zol pre-treatment, no significant increase in the bacterial attachment to discs treated with etidronate was detected (Figure 2A,B). To test whether the nitrogen group has a role in bacterial attachment, we pre-treated the bacteria with an amino acid, histidine, which has the same imidazole ring structure as that in zol molecule. The attachment of histidine-treated bacteria to zol-modified dentin discs was then assessed. Interestingly, the pretreatment of bacteria with histidine decreased the bacterial attachment to zol-modified dentin discs (*p* = 0.0102; Figure 2C). Previous studies have shown that amino acids, such as histidine, glutamate and aspartate are utilized by *F. nucleatum* as a nutrient source [21]. To account for the effect of histidine on the growth of bacteria, bacterial culture media were supplemented with histidine (0 mg/mL, 1 mg/mL, 2 mg/mL) and their growth curve was analyzed. Interestingly, there was an increase in the growth of bacteria with histidine pre-treatment (*p* < 0.001; Figure 2D). Therefore, despite the increase in proliferation, the saturation of bacteria with histidine reduced their affinity for the zol-coated dentin surface, confirming that such affinity may be specific to the nitrogen-containing imidazole ring of the zol molecule.

### 2.4. Localized Delivery of F. nucleatum and A. israelii Causes Osteonecrosis in an Extraction-Type Bone Defect at an Extra-Oral Site Rich in Matrix-Bound Bisphosphonates

Previous studies have shown that MRONJ lesions are heavily colonized with bacteria [1,19], with certain species such as *Actinomyces* and *Fusobacterium* predominating. We tested whether MRONJ-associated bacteria are critical to the development of MRONJ using an extraction-style bone defect in an extra-oral bone site, the proximal tibia. Micro CT, gross, and histologial analyses of the defect revealed that untreated rat defects showed adequate defect healing with or without pre-existing infection (Figure 3A) and zol-treated rats without pre-existing infection showed normal soft tissue and bone healing, comparable to the control animals (Figure 3B). On the other hand, in zol-treated rats, defects pre-infected with *F. nucleatum* and *A. israelii* showed signs of osteolysis, persistent inflammation, periosteal reaction, cortical thickening, and a lack of cortical bridging and bone fill of the defect as shown by micro CT and histological analyses (Figure 3C). Deficient soft tissue healing was observed in three out of eight rats in the zol/*A. israelii* group, exposing underlying necrotic bone similar to findings in MRONJ. Animals from all other groups showed complete soft tissue healing over the defect. A chi-square analysis confirmed that infection caused a significant decrease in cortical bridging in zol-treated animals, while having no effect on the controls; it also confirmed the existence of the synergy between bacterial infection and matrix-bound zol on bone healing in terms of cortical bridging (*p* = 0.0149). A flow cytometry analysis of the popliteal and inguinal lymph nodes that drain the site of tibial defect, showed a decrease in the DC population. The level of expression of CD11c (mean fluorescence intensity (MFI)) was significantly decreased in zol-treated rats compared to the control with *F.nucleatum* and *A.israelii* infection (*p* = 0.00634, *p* = 0.00638, respectively; Figure 3). These results were consistent with our findings in the MRONJ rat model showing impaired DC activity and defective bacterial clearance following surgical extractions in zol-treated rats.

## 3. Discussion

The pathogenesis of MRONJ is multifactorial, with the majority of the occurrences reported following an invasive dental trauma in patients on nitrogen-containing bisphosphonates treatment. The fact that bisphosphonates accumulate in the bone matrix suggests that even stopping the treatment would not eliminate their biological effect. However, little is known about the bioavailability and biological activity of matrix-bound bisphosphonate molecules.

In previous studies, we demonstrated that matrix-bound zoledronate molecules were bio-accessible and bioactive both in vitro and in vivo, playing an important role in predisposing the oral microenvironment to osteonecrosis through their action on monocyte-lineage cells [22,23]. In this study, we tested the hypothesis that matrix-bound bisphosphonates provide a conducive environment for colonization by periodontal pathogens, providing another predisposing factor for osteonecrosis.

We tested the effect of zol on the attachment of different bacterial strains most commonly isolated from MRONJ lesions to mineralized matrices, namely *F. nucleatum*, and *A.israelii* [1,2]. Our results showed that these bacteria consistently favored zol-treated dentin surfaces. Studies by other groups similarly showed the enhanced attachment of *S. aureus* on hydroxyapatite (HA) coated with nitrogen containing bisphosphonate N-BP Pamidronate [12]. Furthermore, the bacterial colonization of HA discs was significantly higher for *Streptococcus mutans*, *Staphylococcus aureus*, and *Pseudomonas aeruginosa* in the presence of bisphosphonates vs. controls [14]. Another study [24] showed that zol promoted the adherence of radiolabeled *Streptococcus mutans* to saliva-coated HA and increased the proliferation of oral bacteria obtained from the plaque of healthy individuals, which was not in line with our results that showed no effect of zol on bacterial proliferation.

We also tested the effect of zol on the attachment capacity of *P. gingivalis*, one of the most common periodontal pathogens. Importantly there was no direct effect of zol treatment on the *P. gingivalis* growth curve. Our results showed a significant increase in the attachment of WT *P. gingivalis* to the zol-treated discs compared with the control discs. Moreover, the *P. gingivalis* fimbrial mutant MFB *FimA**-/Mfa1*- failed to bind to zol-treated discs, suggesting that fimbriae are important for *P. gingivalis* attachment to a zol-treated matrix. Thus, the possibility that zol might directly modulate the fimbriae expression of *P. gingivalis* was conceivable. This led us to test the effect of zol exposure on the gene and protein expression of major and minor fimbriae in *P. gingivalis*. While we found some increase in *fimA* and *mfa-1* gene expression in zol-treated *P. gingivalis*, the differences did not reach statistical significance.

Given that the vast majority of MRONJ occur with nitrogen-containing bisphosphonates, we compared the effect of zol to that of the non-nitrogen-containing etidronate on the bacterial attachment to dentin discs. Our results showed more bacterial attachment in the presence of zol than in the presence of etidronate. Zol was shown to have the highest affinity to HA of all bisphosphonates [25]. We hypothesized that the nitrogen group played a role in rendering zol-treated surfaces more favorable for bacterial attachment. To further investigate this possibility, we pretreated bacteria with histidine, an amino acid that has a nitrogen group similarly structured to that of zol. The pretreatment of *F. nucleatum* with histidine led to its decreased attachment to zol-modified dentin discs. The imidazole structure in the histidine side chain mimics that found in the zol molecule, thus the pretreatment of the bacteria with histidine potentially saturated the imidazole binding sites in the bacteria, decreasing their binding capacity to zol-treated HA. This suggests a potential role for the nitrogen group within the zol molecules in the increased affinity of bacteria to zol-bound bone surfaces.

Amino acid fermentation is considered a primary source of energy for *F. nucleatum*, and it has been shown that glutamate, histidine and aspartate are utilized first [21]. This is consistent with our findings that the direct treatment of bacterial cells with histidine led to an increase in their growth. *F. nucleatum* utilizes amino acids likely through amino acid transporters [21]. However, it is not well understood whether there is a binding site or receptor on the bacterial surface that recognizes histidine. Thus, it is plausible that the imidazole ring in zol actually mimics the molecular structure of histidine, similarly attracting bacterial cells to it. Moreover, previous studies have shown the importance of aromatic rings in the binding of *S. gordonii* to human salivary amylase (HSAmy), as site-directed mutations to those rings led to a marked decrease in its attachment [26].

In our previous studies, we demonstrated that there was an increase in the oral bacterial load in zol-treated rats compared to the control before performing any procedures [23]. Here we tested whether oral bacteria played a localizing role, limiting the osteonecrosis to the oral cavity. We produced extraction-like defects in an extra-oral bone site rich in matrix-bound bisphosphonate to mimic oral cavities in otherwise sterile wound environments. We found that, in zoledronate-treated animals, these defects healed in the same pattern as those in the controls. However, the pre-inoculation of the defect with oral bacteria (*F. nucleatum* or *A. israelii*), induced early signs of osteonecrosis and reduced bone bridging in the zol treated animals compared to the controls. These results suggest that oral bacteria play a synergistic role in the induction of osteonecrosis and provides a potential explanation as to why the condition is limited to the jaws. The data also complement the findings of our previous work [22,23], illustrating a complex, multifactorial induction mechanism for MRONJ.

The hypothesis that oral microbiota play an important role in MRONJ pathogenesis has long been proposed [1,8]. However, whether the bacterial infection is a trigger or an outcome of osteonecrosis remains to be discovered. Previous studies have suggested that periodontitis and associated bacteria could be involved in the development of MRONJ [9]. Clinically, periodontitis has been identified as one of the major MRONJ risk factors [5].

Our studies focused only on zol since it carries the highest risk of ONJ. Other studies, using pamidronate-coated surfaces, have shown a similar increase in the bacterial attachment to HA, and their authors proposed that this increased attachment is mediated through direct electrostatic forces between bacterial adhesins and the cationic amino group in pamidronate [13]. Another mechanism proposed was that the cationic amino group in pamidronate acts as a mimic to host proteins that provides a substrate for binding to microbial surface components recognizing the adhesive matrix molecules (MSCRAMS) of bacterial cells [12].

Our findings are consistent with another study showing that the inoculation of *F. nucleatum* into an extraction site in mice treated with pamidronate exacerbated osteonecrosis [10]. However, a limitation of the cited study was the presence of a complex microbiota in the oral cavity, which increases the variables and makes interpretation difficult.

*Actinomyces* infections have frequently been detected at MRONJ sites in human patients [1,2]. However, it is unclear whether their presence was commensal or pathogenic, and whether it contributed to causing osteonecrosis or occurred afterwards as a consequence of having exposed necrotic bone [27]. Multiple theories have been postulated regarding the role of the oral microbiome in the development of MRONJ. Some investigators suggested that MRONJ is a result of bacterial-induced bone resorption [1,3,4]. Others have proposed that osteonecrosis occurs through a two-stage process [5]. In the first stage, periodontitis initiates periodontal tissue inflammation, the production of inflammatory cytokines, and increased the expression of RANKL. This results in further tissue destruction, impaired angiogenesis, and infection-induced bone resorption. In the second stage, osteoclastic activity is impaired by the direct effect of the drug, leading to the failure of alveolar bone removal. This subjects the alveolar bone to the continuous exposure to inflammatory cytokines, bacterial toxins and oxidative stress, which are highly toxic to bone cells and lead to necrosis [7,8]. Moreover, the impairment of bone resorption makes it difficult for adaptive immune cells to gain access to the infection site and allows the infection to persist [8].

This theory was supported by a cross-sectional study that compared cases with MRONJ history to controls with no MRONJ development, all of whom received BP treatment. Periodontitis-associated bacteria were suggested as potential etiologic factors in MRONJ pathogenesis [9]. Data from animal models of periodontitis supported this theory as well [10], whereby nitrogen-containing bisphosphonates induced ONJ-like lesions in rice rat models of periodontitis [5], and osteonecrosis was observed in ligature-induced periodontitis mouse models in zol-treated animals [7]. Moreover, a histological examination of experimental bone defects in the mandibles versus femurs of Wistar rats, previously exposed to zol or saline, showed larger necrotic areas in the zol groups associated with *A. actinomycetemcomitans* infection to both sites compared to the controls [11].

The current study supported previous postulations that the accumulation of BP in the bone matrix favored bacterial attachment, contributing to the pathogenesis of MRONJ [12,13,14]. The attachment of gram-positive strains, such as *actinomyces*, would facilitate the adherence of other microbial species, forming a resistant biofilm and causing dysbiosis [13,14]. Whether by invasive trauma or periodontitis, exposing the afflicted bone surfaces would promote the further bacterial colonization of alveolar bone [14,15]. Other factors may also contribute to making the alveolar bone uniquely vulnerable to osteonecrosis, including the effect of the drug on the oral immune response [23] and the unique nature of invasive dental trauma, which exposes the bone to the oral environment during the healing process. The accumulation of nitrogen-containing bisphosphonates in the bone matrix ensures that the alveolar bone continues to be vulnerable to osteonecrosis even after stopping the treatment—unlike other antiresorptive medications. We have recently found promising results using locally delivered chelating agents such as EDTA, which improved socket healing and prevented osteonecrosis in zol-treated animals. [22] Another study showed that the displacement of pre-adsorbed nitrogen-containing bisphosphonates using a weaker BP significantly attenuated osteonecrosis [28]. Thus, future research should be focused on therapeutic approaches that locally remove matrix bound bisphosphonates from the bone surface.

## 4. Materials and Methods

### 4.1. Bacterial Strains and Culture

Wild-type *Porphorymonas gingivalis* (*P. gingivalis* 381), its isogenic double-fimbriae mutant (MFB*FimA*-/*Mfa1*-), which lacks both major and minor fimbriae, *Fusobacterium nucleatum* (*F. nucleatum*; ATCC 49256) and *Actinomyces isrealii* (*A. isrealii*; ATCC 12103) were maintained anaerobically (10% H_2_, 10% CO_2_ and 80% N_2_) in a Coy Laboratory vinyl anaerobic system glovebox (model 1025/1029) at 37 °C. *P.gingivalis* and *F. nucleatum* were grown in Wilkins-Chalgren anaerobe broth and *A. isrealii* were first grown in Muller Hinton Agar plates with 5% sheep blood (BD Biosciences, Franklin Lakes, NJ, USA) then transferred to BBL Actinomyces broth (BD Biosciences, Franklin Lakes, NJ, USA). All bacterial suspensions were washed three times in PBS prior to re-suspension for spectrophotometer density reading. *P. gingivalis* cultures OD660 of 0.1 was previously determined to be equal to 5 × 10^7^ CFU, *F. nucleatum* OD600 of 0.1 equal to 75 × 10^6^ CFU and, finally, *A.isrealii* OD600 of 0.1 equal to 10^8^ CFU. All bacterial cultures were diluted to provide 10^8^ CFU.

### 4.2. Zol Coating of Dentin Discs

To confirm the coating of dentin discs with zol, dentin discs were treated with varying doses (0, 0.1 µM, 1 µM) of AF647 zol (Biovinc, LLC, Pasadena, CA, USA) in a 24-well plate for 24 h. The discs were then washed thoroughly with sterile phosphate–buffered saline (PBS) three times. Then discs were imaged using an Ami-X optical fluorescence imaging system (Spectral Instruments Imaging, LLC, Tucson, AZ, USA) with excitation and emission indocyanine green (ICG) filters. The fluorescent images were taken at the 640 nm and 670 nm excitation and emission wavelengths, respectively.

### 4.3. Assessment of Attachment of Oral Bacteria to Dentin Discs

Zol (0 µM, 0.1 µM, 1 µM, 10 µM, 100 µM) in 500 μL PBS was added to sterile dentin discs in a 12-well plate and incubated at room temperature for 24 h, followed by washing thoroughly three times with PBS. Discs were then infected with 108 CFU of different bacterial strains; *P. gingivalis* 381, MFB *FimA*-/*Mfa1*-, *F. nucleatum* (ATCC 49256) and *A. isrealii* (ATCC 12103) were grown at their mid-log phase and incubated anaerobically for 24 h. Discs with no infection or with the respective drug only were included in the study as negative controls. The discs were then washed thoroughly in PBS with shaking to get rid of unattached bacteria, followed by staining with a live/dead bacterial stain (Thermo Fisher Scientific, West Columbia, SC, USA), with the live bacteria with intact membrane stained green (SYTO9 stain) and the dead stained red (propidium iodide). The presence of attached bacterial cells was visualized using a 780 multiphoton confocal microscope with a 20× dipping lens (Zeiss AxioIma, Carl Zeiss Microscopy GmbH, Jena, Germany). Images were analyzed with Image J software using plug-in macros with a set threshold for fluorescent intensity based on the negative control. To determine whether that increase in attachment was due to a zol-induced increase in bacterial proliferation or due to increased attachment per se, we tested the effect of the zol direct treatment on bacterial growth. Zol (10 µM and 100 µM) was added to growing bacterial cultures and growth was assessed with OD660 nm readings.

### 4.4. Effect of Histidine on Bacterial Growth and Attachment to Zol-Coated Dentin Discs

To determine the effect of histidine on the growth of bacteria, we supplemented bacterial culture with histidine (Sigma Aldrich, St Louise, MO, USA) at concentrations of 0 mg/mL, 1 mg/mL and 2 mg/mL. Growth of bacteria was then assessed after 3, 6, 12 and 24 h using a spectrophotometer reading at OD600 (BioMate UV-Visible spectrophotometer, Thermofisher Scientific, West Columbia, SC, USA). To determine effect of histidine on bacterial attachment to zol-modified dentin discs, *F. nucleatum* were treated with histidine (2 mg/mL). After 24-h incubation, 10^8^ CFU of histidine-treated bacteria were used to infect dentin discs pretreated with either zol or PBS. Then their attachment was assessed as described above.

### 4.5. rtPCR for Relative Gene Expression Analysis of Minor (mfa-1) and Major (fimA) Fimbriae in P. gingivalis Treated with Zol

Quantitative real-time PCR (qPCR) was performed using an iTAQ universal probe supermix (BIO-RAD, Heculeus, CA, USA). Individual TaqMan^®^ gene expression primers (Applied Biosystem, Foster city, CA, USA) were designed to target the mRNA of 16S rRNA of *P. gingivalis* (PG16S assay AIY9ZZ2; GenBank: AB035455.1), *fimA* (FIMA assay AIY9ZZ0) and *mfa1* (MFA1 AIX01UM). RT-PCR was run in a StepOnePlus™ Real-Time PCR System. Gene expression of minor (mfa-1) and major (fimA) fimbriae were quantified using 16S rRNA as housekeeping. Gene expression of minor (*mfa-1*) and major (*fimA*) fimbriae were quantified using 16S rRNA as housekeeping. For calculations and statistical analysis, fold changes were calculated using the (2^−ΔΔCT^) method in the experimental samples. Statistical analysis for gene expression was performed using the one-sample *t*-test, which estimates the calculated difference (in fold regulation) between experimental and control samples. A *p*-value of < 0.05 was used as the cut-off for significant difference.

### 4.6. Animals and Surgical Procedures

Thirty female Sprague-Dawley rats (aged 10–12 months) were treated with either intravenous zol 80 μg/kg/week (experimental, *n* = 15) or saline (control, *n* = 15) for 9 weeks, as previously described [18]. Animals were then assigned randomly into 6 groups as follows (*n* = 5 per group):

Group 1: Control rats with no infection (sham injection)

Group 2: Control rats with *F. nucleatum* local infection

Group 3: Control rats with *A. israelii* local infection

Group 4: Zol-treated rats with no infection (sham injection)

Group 5: Zol treated rats with *F. nucleatum* local infection

Group 6: Zol treated rats with *A. israelii* local infection

Subcutaneous injection of *F. nucleatum* and *A. israelii* in 200 uL PBS, 2% carboxy methyl cellulose (CMC) or CMC alone (vehicle) were delivered to the proximal tibiae unilaterally at day −1 prior to the surgery day. At day 0, an incision was made using a no. 15 blade, and a unilateral extraction-like 1-mm circular bone defect was created in the anteromedial cortex using a round 2 mm carbide bur (Henry Schein, Melville, NY, USA) with irrigation, and then incision was sutured with 3-0 silk (Henry Schein, Melville, NY, USA). Surgery was done using strictly aseptic procedures. After suturing, 10^9^ CFU of *F. nucleatum* or *A. israelii* in CMC or CMC alone (*n* = 5 per group) were injected into the site of the defect. Local delivery of 10^9^ CFU bacterial cells was repeated every other day for a week. Then rats were allowed to heal for 2 more weeks, after which they were euthanized, clinical pictures were taken, and tibiae were harvested and fixed for histomorphometric analysis.

### 4.7. Flow Cytometry

Popliteal and inguinal lymph nodes draining the tibial defect sites were collected, and were processed for flow cytometry analysis of dendritic cell population. Staining was done on ice in a Flow Cytometry Staining Buffer (Affymetrix, eBioscience, Thermofisher scientific, Waltham, MA, USA). Blocking of FC receptor (FCR) using mouse FcR blocking reagent (MACS/Miltenyi Biotec Inc., San Diego, CA, USA) was conducted for 15 min on ice and protected from light; this was followed by adding the fluorophore-conjugated antibody (at the recommended concentration) on ice for 30 min, after which cells were washed and re-suspended in a flow staining buffer. Flow data were acquired using the BD FACSDIVA software (BD Bioscience, Franklin Lakes, NJ, USA) on a FACSCanto (BD bioscience). Flow data were analyzed by Flowjo software, evaluating: the 10.2. flow cytometry antibodies used; the anti-mouse CD11c PerCp-cyanine5.5, PE; the clone N418; and the MHC class II (I-A/I-E) (Affymetrix, eBioscience, Thermofisher scientific, Waltham, MA, USA).

### 4.8. Micro CT Analysis

Rat tibiae were scanned using the 1172 Skyscan System (Bruker, Belgium) at 12 µm image pixel size with a 0.5 mm Al filter, 0.2 rotation step and averaging frames of 3. NRecon software (Skyscan) was used for 3-D reconstruction and the 3-D analysis of alveolar bone healing in the maxillary first molar was done using CTAn software (Skyscan). Reconstructed images were analyzed for signs of osteomyelitis, periosteal thickening and bridging of the bone defect.

### 4.9. Histology and Histomorphometry

Tibia samples were embedded in paraffin and 5 µm-thick sagittal sections were made through the defect site, then stained with Hematoxylin and Eosin. Three random microphotographs were taken at different magnifications of the defect site (Zeiss AxioIma, Carl Zeiss Microscopy GmbH, Jena, Germany). The ROI was examined for new bone formation, persistent inflammation, tissue destruction, bone necrosis, and sub-periosteal reactive bone formation. Cortical bridging was analyzed as a dichotomous variable (y/n) and groups were compared using the chi-square test.

### 4.10. Statistical Analysis

Statistical analysis was done using the Graph Pad Prism software version 6 (Graph Pad Software, La Jolla, CA, USA). In-vitro experiments were repeated at least three times. Data values were reported as means ± SD. Normality assumption was evaluated when applicable, using the Shapiro-Wilk test. When normality assumptions were not met, an alternative non-parametric test was done. A one-way ANOVA test with significance defined as *p* < 0.05, a confidence level of 95% confidence interval and a Bonferroni post-hoc comparison was used to compare multiple groups. The unpaired T-test was used to compare two groups in the in vivo experiments. Chi-square analysis was performed to examine the role of bacterial infection and zol on bone healing.

## 5. Conclusions

Matrix-bound zol enhances the oral biofilm colonization of hydroxyapatite. This enhancement depended on the presence of the nitrogen-containing group in bisphosphonates. Oral biofilm rendered extra-oral bone susceptible to medication-related osteonecrosis, suggesting that it has an important role in the induction of MRONJ.

## Figures and Tables

**Figure 1 antibiotics-10-01380-f001:**
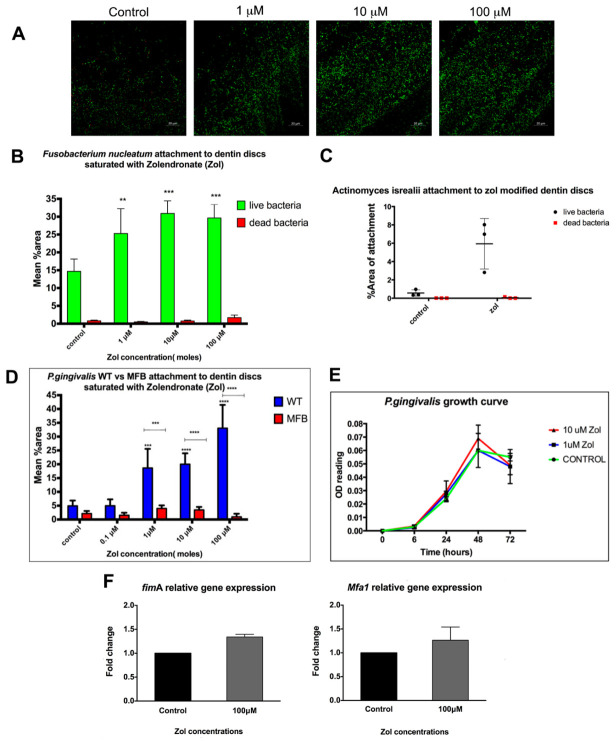
Effect of matrix-bound zol on the attachment of different bacterial strains. (**A**) Confocal microscopy images of *F. nucleatum* stained with a fluorescent live (green) and dead (red) stain on zol-coated dentin discs. (**B**) Summary bar graph of the quantitative analysis of *F. nucleatum*’s attachment to zol-coated dentin discs. (**C**) Summary bar graph showing *A. Israelii*’s attachment to zol-modified dentin discs. (**D**) Mean % area attachment of WT *P. gingivalis* vs. MFB (*FimA^−^/Mfa1^−^*) attachment on zol-coated dentin discs. (**E**) Growth curve showing *P. gingivalis* proliferation with zol treatment. (**F**) An rtPCR analysis showing the relative mRNA expression of the major (*fimA*) and minor (*mfa1*) fimbriae in *P. gingivalis* with and without zol treatment. (Data represent mean ± SD of three independent experiments ** *p* < 0.01, *** *p* < 0.001, **** *p* < 0.0001).

**Figure 2 antibiotics-10-01380-f002:**
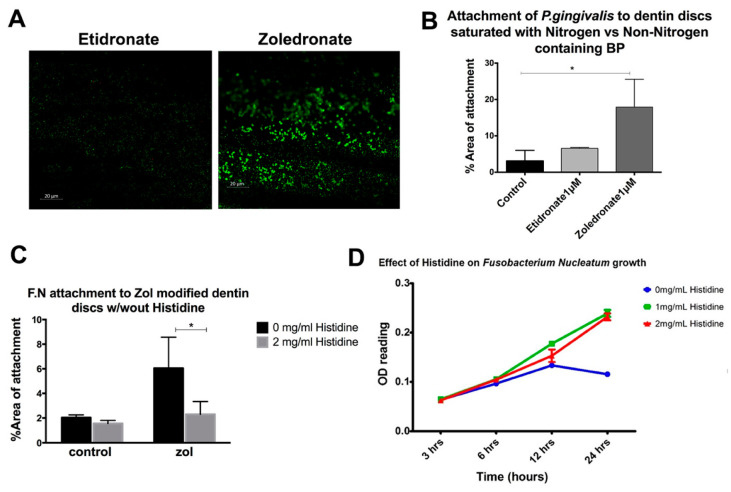
Role of the nitrogen group in bisphosphonate in enhancing the bacterial attachment to a mineralized matrix. (**A**,**B**) Quantitative analysis of the attachment of *P. gingivalis* pretreated with histidine on zol- and etidronate-coated dentin discs. (**C**) Attachment of *F. nucleatum* (FN) to zol-coated dentin discs with and without the histidine pre-treatment. (**D**) Growth curve of *F. nucleatum* with the histidine treatment (0 mg/mL, 1 mg/mL, 2 mg/mL) (* *p* < 0.05).

**Figure 3 antibiotics-10-01380-f003:**
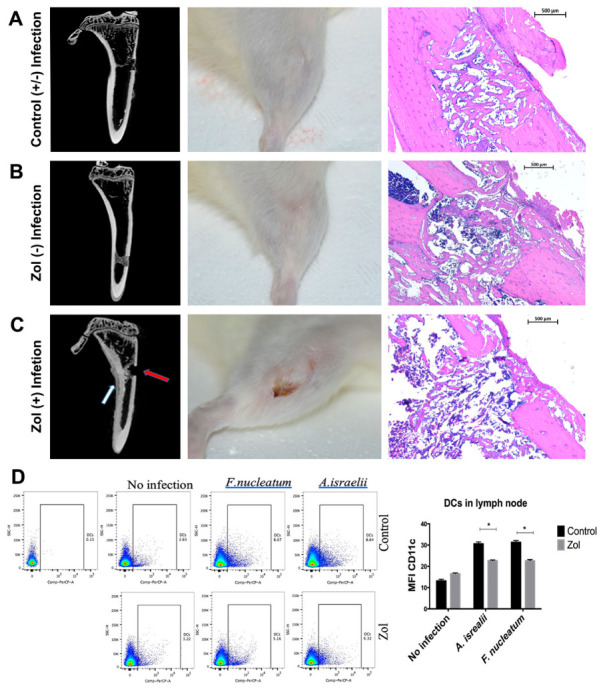
Development of osteonecrosis in a tibial model requires both zol treatment and the pre-existence of oral bacteria. (**A**) A gtibial defect in untreated (control) rat tibia with or without pre-existing local infection with *A. israelii*: left panel—a micro CT showing signs of normal wound healing and cortical bridging; middle panel—a clinical picture showing complete soft tissue healing; right panel—a representative histological photomicrograph showing adequate defect healing and bone fill. (**B**) A tibial defect in zol-treated rats without pre-existing *A. israelii* local infection: left panel—a micro CT showing normal gap bridging and callus filling with no signs of a periosteal reaction or cortical thickening, comparable to the control rats; middle panel—a clinical picture showing complete soft tissue healing; right panel—a representative histological photomicrograph showing adequate defect healing and bone fill. (**C**) A tibial defect in zol-treated rat tibia with pre-existing infection: left panel—a micro CT showing signs of osteolysis, a lack of cortical bridging (red arrow) and an inflammatory periosteal reaction (white arrow); middle panel—a clinical image showing delayed wound healing and a persistent dehiscence, exposing the underlying necrotic bone surfaces, similar to MRONJ; right panel—a representative histological photomicrograph showing adequate defect healing and bone fill. (**D**) A flow cytometry scattergram and corresponding summary graph showing the MFI in the DC population in the popliteal and inguinal lymph nodes in the control (upper panel) and zol-treated (lower panel) rats with *F. nucleatum*, *A. israelii* and no infection. Statistical analysis done by multiple T test (* *p* < 0.05).

## Data Availability

The data presented in this study are available on request from the corresponding author.

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
