# Peer review of "Matrix-Bound Zolzoledronate Enhances the Biofilm Colonization of Hydroxyapatite: Effects on Osteonecrosis"

_antibiotics, 2021, doi:10.3390/antibiotics10111380_

Round 1

Reviewer 1 Report

Dear authors

Below you will find the comments and observations of your manuscript

  • Scientific names must be in italics
  • In the photos you must include measurement scale (mm, um, nm ...)

Regards

Reviewer

Author Response

Thank you for your comments. All scientific names are now in Italic. We added the scale bars to the original microscope images using the calibrated microscope software. 

Reviewer 2 Report

Interesting topic, very well documented, organised research, useful for the clinical treatment, especially for the prevention of MRONJ.

I suggest to add some more clinical treatment suggestions related with your study, the clinical utility of your research.

Author Response

Thank you for your kind comments. We added a paragraph at the end of the discussion to point to our ongoing studies, which address exactly the clinical applicability of this study.

Reviewer 3 Report

I would like to congratulate the authors on their interesting publication. In the future, I recommend to do a new study that would include denosumab comparison with zolendonate of bacterial colonization.

Author Response

Thank you for your kind comment. Such a study is important. We have been trying to conduct it for some time but ran into some logistical problems. It is one of our priorities in the near future.